# Bag of Image Patch Embedding
# Behind the Success of Self-Supervised Learning

**Yubei Chen** [5,*]                                                      *ybchen@ucdavis.edu*

**Adrien Bardes** [1,2,*]

**Zengyi Li** [3]

**Yann LeCun** [1,4]

[1]Meta AI - FAIR [2] Inria, École normale supérieure, CNRS, PSL Research University
[3]Aizip, Inc. (Work performed while at Redwood Center, UC Berkeley)
[4]Center for Data Science, New York University
[5]ECE Dept., UC Davis (Work performed while at Meta AI - FAIR and CDS, NYU)
[*]Equal contribution

Reviewed on OpenReview: `https://openreview.net/forum?id=r06xREo3QG`

## Abstract

Self-supervised learning (SSL) has recently achieved tremendous empirical advancements in learning image representation. However, our understanding of the principle behind learning such a representation is still limited. This work shows that joint-embedding SSL approaches learn a representation of image patches, which reflects their co-occurrence. Such a connection to co-occurrence modeling can be established formally, and it supplements the prevailing invariance perspective. We empirically show that learning a representation for fixed-scale patches and aggregating local patch representations as the image representation achieves similar or even better results than the baseline methods. We denote this process as *BagSSL*. Even with $32 \times 32$ patch representation, BagSSL achieves 62% top-1 linear probing accuracy on ImageNet. On the other hand, with a multi-scale pretrained model, we show that the whole image embedding is approximately the average of local patch embeddings. While the SSL representation is relatively invariant at the global scale, we show that locality is preserved when we zoom into local patch-level representation. Further, we show that patch representation aggregation can improve various SOTA baseline methods by a large margin. The patch representation is considerably easier to understand, and this work makes a step to demystify self-supervised representation learning.

## 1 Introduction

In many application domains, self-supervised representation learning experienced tremendous advancements in the past few years. In terms of the quality of learned features, unsupervised learning has caught up with supervised learning or even surpassed the latter in many cases. This trend promises unparalleled scalability for data-driven machine learning in the future. One of the most successful self-supervised learning paradigms is joint-embedding SSL (Wu et al., 2018; Chen et al., 2020a; Grill et al., 2020; Chen et al., 2020b; Bardes et al., 2021; Zbontar et al., 2021; Yeh et al., 2022; Li et al., 2022), which uses a Siamese network architecture (Bromley et al., 1993). These methods follow a general goal: to transform different views (augmentation) of the same instance (image) closer in the representation space, and meanwhile, the new space is not collapsed, or in other words, important geometric and stochastic structures are preserved.

While we celebrate the empirical success of SSL, our understanding of this learning process still needs to be improved. This work shows that ***joint-embedding SSL approaches are learning a representation***

***of image patches, which reflects their co-occurrence.*** To demonstrate this, we first establish a formal connection between joint-embedding SSL and co-occurrence modeling. Then, we show that learning a representation for fixed-scale patches and linearly aggregating patch representations (bag-of-local-features) as the image representation achieves similar or even better results than learning the representation with multi-scale crops. Empirical results are shown with several SSL methods(Chen et al., 2020a; Bardes et al., 2021; Li et al., 2022; Grill et al., 2020) on CIFAR10, CIFAR100, ImageNet100, and ImageNet-1K datasets. Even with $32 \times 32$ patch representation, we can achieve 62% top-1 linear probing accuracy on ImageNet-1K. And KNN classifier also works surprisingly well with the aggregated patch feature. These findings resonate with recent works in supervised learning based on local features (Brendel & Bethge, 2018; Dosovitskiy et al., 2021; Trockman & Kolter, 2023). We further show that for baseline SSL methods pretrained with multi-scale crops, the whole-image representation is approximately a linear aggregation of local patch embeddings. The local patch representation is considerably easier to understand, and we provide visualization to show that the representation space preserves locality when we zoom into the local patch representation. These discoveries supplement the prevailing invariance perspective, provide a useful understanding of the success of joint-embedding SSL, and make a step to demystify self-supervised representation learning.

## 2 Related Works

**Joint-embedding SSL: Invariance without Collapse.** Joint-embedding SSL (or instance-based SSL) (Wu et al., 2018) views each of the images as a different class and uses data augmentation (Dosovitskiy et al., 2016) to generate different views from the same image. As the number of classes equals the number of images, it is formulated as a massive classification problem, which may require a huge buffer or memory bank. Later, SimCLR (Chen et al., 2020a) simplifies the technique significantly and uses an InfoNCE-based formulation to restrict the classification within an individual batch. At the same time, it's widely perceived that contrastive learning needs the "bag of tricks," e.g., large batches, hyperparameter tuning, momentum encoding, memory queues, etc. Later works (Chen & He, 2021; Yeh et al., 2022; HaoChen et al., 2021) show that many of these issues can be easily fixed. Recently, several even simpler non-contrastive learning methods(Bardes et al., 2021; Zbontar et al., 2021; Li et al., 2022) are proposed, where one directly pushes the representation of different views from the same instance closer while maintaining a non-collapsing representation space. Joint-embedding SSL methods mostly differ in their means of achieving a non-collapsing solution. This include classification versus negative samples(Chen et al., 2020a), Siamese networks (He et al., 2020; Grill et al., 2020) and more recently, covariance regularization (Ermolov et al., 2021; Zbontar et al., 2021; Bardes et al., 2021; HaoChen et al., 2021; Li et al., 2022; Bardes et al., 2022). The covariance regularization has also long been used in many classical unsupervised learning methods (Roweis & Saul, 2000; Tenenbaum et al., 2000; Wiskott & Sejnowski, 2002; Chen et al., 2018), also to enforce a non-collapsing solution. There is a duality between the spectral contrastive loss(HaoChen et al., 2021) and the non-contrastive loss, which we briefly discuss the intuition in Appendix B.

All previously mentioned joint-embedding SSL methods pull together representations of different views of the same instance. Intuitively, the representation would eventually be invariant to the transformation that generates those views. We would like to provide further insight into this learning process: The learning objective can be understood as using the inner product to capture the co-occurrence statistics of local image patches. We also provide visualization to study whether the learned representation truly has this invariance property.

**Patch-Based Representation.** Many works have explored the effectiveness of path-based image features. In the supervised setting, Bagnet(Brendel & Bethge, 2018) and Thiry et al. (2021) showed that aggregation of patch-based features could achieve most of the performance of supervised learning on image datasets. In the unsupervised setting, Gidaris et al. (2020) performs SSL by requiring a bag-of-patches representation to be invariant between different views. Due to architectural constraints, vision transformer-based methods also naturally use a patch-based representation (He et al., 2022; Bao et al.).

**Learning Representation by Modeling the Co-Occurrence Statistics.** The use of word vector representation has a long history in NLP, which dates back to the 80s (Rumelhart et al., 1986; Dumais, 2004). Perhaps one of the most famous word embedding results, the word vector arithmetic operation,

was introduced in Mikolov et al. (2013a). Particularly, to learn this embedding, a task called "skip-gram" was used, where one uses the latent embedding of a word to predict the latent embedding of the word vectors in a context. A refinement was proposed in Mikolov et al. (2013b), where a simplified variant of Noise Contrastive Estimation (NCE) was introduced for training the "Skip-gram" model. The task and loss are deeply connected to the SimCLR and its InfoNCE loss. Later, a matrix factorization formulation was proposed in Pennington et al. (2014), which uses a carefully reprocessed concurrence matrix compared to latent semantic analysis. While the task in Word2Vec and SimCLR is apparently similar, the underlying interpretations are quite different. In joint-embedding SSL methods, one pervasive perception is that the encoding network is trying to build invariance, i.e., different views of the same instance shall be mapped to the same latent embedding. This work supplements this classical opinion and shows that similar to Word2Vec, joint-embedding SSL methods can be understood as building a distributed representation of image patches by modeling the co-occurrence statistics.

## 3 Joint-Embedding SSL Models Image Patch Co-Occurrence

As we discussed earlier, recent works show that there exists a duality Garrido et al. (2023) between contrastive SSL and non-contrastive SSL methods, and these methods are essentially solving the same problem. So we only need to establish the connection between co-occurrence statistics modeling and one joint-embedding SSL method. In the following, we show that the spectral contrastive learning HaoChen et al. (2021) loss function models the co-occurrence statistics. We also replace multi-scale crop augmentation with fixed-scale image patches during SSL training. Thus the learning process is precisely modeling the patch-level co-occurrence statistics. After training, SSL has learned a representation for fixed-scale image patches. The whole-image representation is a linear aggregation of local patch representations.

**Joint-embedding SSL and Co-Occurrence.** Let's assume $\vec{x}_1$ and $\vec{x}_2$ are two color-augmented patches sampled from the dataset. We denote their marginal distribution by $p(\vec{x}_1)$ and $p(\vec{x}_2)$, which includes variation due to sampling different images, spatial locations within an image, and random color augmentation. We denote their joint distribution by $p(\vec{x}_1, \vec{x}_2)$, which is the probability that they co-occur within the same image. $\vec{z}_1$ and $\vec{z}_2$ are corresponding embedding vectors of $\vec{x}_1$ and $\vec{x}_2$ in the representation space. The representation transform is parameterized by a neural network. The next proposition shows that the spectral contrastive loss function is equivalent to co-occurrence statistics modeling.

**Proposition 3.1.** *The spectral contrastive loss function $L_S$:*

$$L_S = \mathbb{E}_{p(\vec{x}_1, \vec{x}_2)} \left[ -\vec{z}_1^T \vec{z}_2 \right] + \lambda \mathbb{E}_{p(\vec{x}_1)p(\vec{x}_2)} \left( \vec{z}_1^T \vec{z}_2 \right)^2 \tag{1}$$

*is equivalent to the co-occurrence statistics modeling loss function $L_C$:*

$$L_C = \int p(\vec{x}_1)p(\vec{x}_2) \left[ w \vec{z}_1^T \vec{z}_2 - \frac{p(\vec{x}_1, \vec{x}_2)}{p(\vec{x}_1)p(\vec{x}_2)} \right]^2 d\vec{x}_1 d\vec{x}_2 \tag{2}$$

*where $\lambda = \frac{w}{2}$, and $w$ is a constant weighting factor.*

The proof is rather straightforward and is presented in Appendix A. The first term in $L_S$ ensures that co-occurring image patches are embedded closer in the embedding space, and the second term in $L_S$ uniformly pushes the patch embeddings away from each other so that the representation space does not collapse.

**Bag-of-Local-Feature Evaluation.** After we have learned a representation of fix-scale image patches, we can embed all of the image patches $\{\vec{x}_{11}, \ldots, \vec{x}_{HW}\}$ within an image into the embedding space. Then, we can obtain the representation $R_{img}$ for the whole image by linearly aggregating (averaging) all patch embeddings. The pipeline is shown in Figure 1. $\vec{z} = g(\vec{h}; \psi)$ and $\vec{h} = f(\vec{x}; \theta)$. We call $\vec{h}$ the *embedding* and $\vec{z}$ the *projection* of an image patch, $\vec{x}$. $\{\vec{x}\}$ are fixed-scale. The function $f(\cdot; \theta)$ is a deep neural network with parameters $\theta$, and $g$ is typically a much simpler neural network with only one or a few fully connected layers and parameters $\psi$. Following a convention in joint-embedding SSL, we use $\vec{h}$ as patch representation rather than $\vec{z}$ for slightly better accuracy, i.e., $R_{img} = \text{mean}_{HW}(\vec{h}_{11}, \cdots, \vec{h}_{HW})$. During SSL training, given an image patch $\vec{x}_i$, the objective tries to pull its projection $\vec{z}_i$ closer to the projections of other co-occurring image patches and to push away non-co-occurring image patches' projections. For easier reference, we denote this process as *BagSSL*.

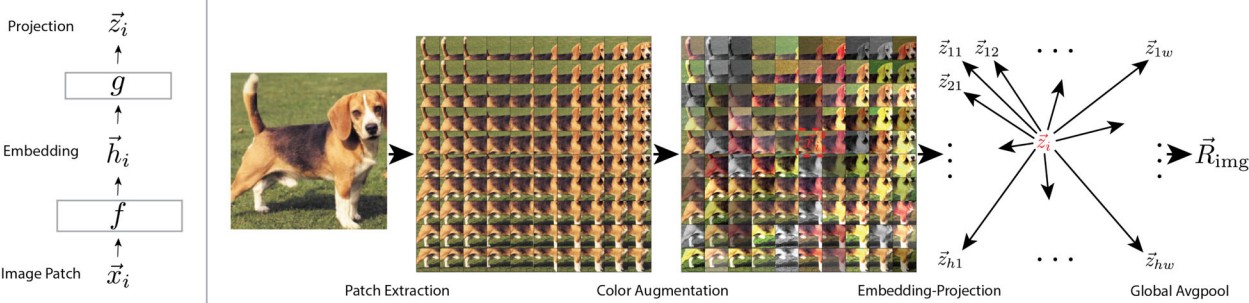

Figure 1: **BagSSL pipeline.** From each image, fixed-size image patches are extracted, color-augmented, and encoded to embedding and projection space. During training, patch projections of co-occurring patches (from the same image) are pulled together while an anti-collapse regularization is applied to push non-co-occurring patch projections away. After training, patch embeddings $\{\vec{h}\}$ from the same image are averaged to form the image representation $\vec{R}_{img}$.

## 4 Empirical Results

In this section, we present a series of empirical results and test BagSSL with several representative joint-embedding SSL methods (SimCLR(Chen et al., 2020a), VICReg(Bardes et al., 2021), TCR(Li et al., 2022), and BYOL(Grill et al., 2020)) on four standard benchmarks (CIFAR10, CIFAR100, ImageNet-100, and ImageNet-1K). Through experiments, we have the following observations:

- BagSSL achieves similar or even better results than baseline methods. Even with $32 \times 32$ patch representation, BagSSL achieves 62% top-1 linear probing accuracy on ImageNet.

- The aggregated patch embedding approximately converges to the whole-image representation in baseline SSL methods with respect to the number of patches aggregated.

- Through visualization, we show that the smaller-scale patch representation preserves locality better.

- We can leverage BagSSL to further improve the representation quality of the baseline methods.

The first two observations show from two different angles that a patch-based SSL representation can potentially explains the performance of the baseline methods. The third observation shows how patch representation supplements the prevailing invariance perspective. And the fourth observation provides practical benefits in addition to understanding.

### 4.1 BagSSL Versus Baseline Methods

We first test BagSSL with four baseline methods. The only difference between BagSSL and the baselines during training is that we replace multi-scale cropping augmentation with fixed-scale image patches. Similarly, two image patches are randomly selected from each image in the batch. While we mainly present linear probing accuracy, Figure 2 shows that k-NN evaluation is consistent with linear probing evaluation. During the evaluation, we show standard central crop evaluation results for both BagSSL and baselines. Further, we show the multi-patch aggregated evaluation and multi-crop aggregated evaluation. In the tables, "patch" means that fixed-scale patch embeddings are aggregated, and "crop" means that multi-scale crop embeddings are aggregated. All implementation details can be found in Appendix C. The main results are highlighted in Table 1, Table 2, Table 3, where BagSSL matches the baselines or surpass them on CIFAR10, CIFAR100, and ImageNet-100 datasets. In Figure 3, we show the results on ImageNet-1K. Interestingly, with $32 \times 32$ patch representation, BagSSL achieves 62% top-1 linear probing accuracy. There is a 5.3% performance gap between BagSSL and VICReg on ImageNet-1K, possibly due to the engineering issue.

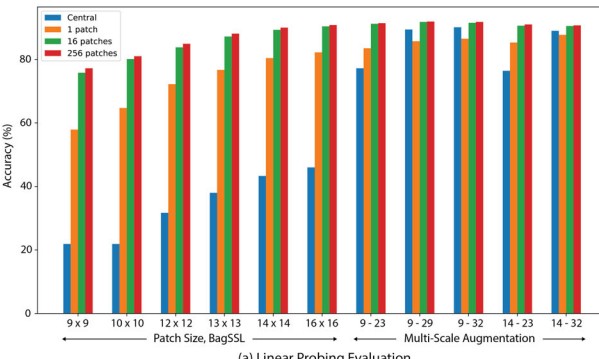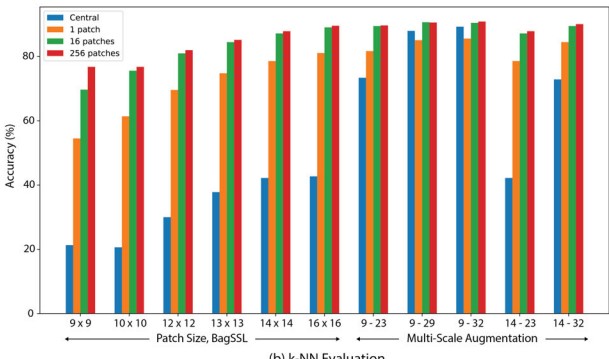

Figure 2: **Linear probing and kNN evaluation on CIFAR-10 are consistent.** We evaluate the performance of a linear classifier (a) and a k-NN classifier (b) for pretraining with various patch sizes and various evaluation setups. During pretraining, BagSSL uses fixed-scale patches, and baseline methods use multi-scale augmentation `RandomResizedCrop(min_scale, max_scale)`. For example, `RandomResizedCrop(0.08, 1.0)` corresponds to randomly and uniformly select crops ranging from $9 \times 9$ to $32 \times 32$. The patch size and augmentation crop size range are marked in the figure. The "Central" evaluation is the standard evaluation protocol where the classifier is trained and evaluated on single fixed central patches of the image, which is the entire image for CIFAR-10. For the $n$ patch evaluation, the classifier is trained and evaluated on the linearly-aggregated embedding of $n$ patches, sampled with the same scale factor as during pretraining. Please note that the "central" evaluation is expected to perform poorly on fix-scale pretraining as the model has never seen the entire image during pretraining.

Table 1: **Performance on CIFAR-10 with patch-based and standard multi-Scale SSL pretraining.** We evaluate the performance of a linear classifier with BagSSL and baseline SSL methods. BagSSL uses *patch-based training*, where $14 \times 14$ image patches are sampled during pretraining. Baseline methods use *multi-scale training*, where the patch scale is uniformly sampled between scale 0.08 and 1.0, which means that multi-scale crops are sampled uniformly from $9 \times 9$ to $32 \times 32$ patches.

| Method | Patch-based training evaluation | | | | Multi-scale training evaluation | | | |
|---|---|---|---|---|---|---|---|---|
| | Central crop | 1 patch | 16 patches | 256 patches | Central crop | 1 crop | 16 crops | 256 crops |
| SimCLR | 46.2 | 82.1 | 90.5 | 90.8 | 90.2 | 86.4 | 91.6 | 91.8 |
| TCR | 46.0 | 82.2 | 90.4 | 90.8 | 90.1 | 86.5 | 91.5 | 91.8 |
| VICReg | 47.1 | 83.1 | 90.9 | 91.2 | 90.7 | 87.3 | 91.9 | 92.0 |
| BYOL | 47.3 | 83.6 | 91.3 | 91.5 | 90.9 | 87.8 | 92.3 | 92.4 |

**CIFAR.** We first provide experimental results on the standard CIFAR-10 and CIFAR-100 datasets (Krizhevsky et al., 2009) using ResNet-34. During training, fixed-scale image patches are used for BagSSL, and baseline methods RandomResizedCrop(0.08, 1.0) augmentation, which means that we randomly choose to sample from $9 \times 9$ to $32 \times 32$ patches as the crops. Sampled patches and crops are upsampled to a uniform $32 \times 32$ resolution before embedding by ResNet-34. The standard evaluation method generates the embedding using the entire image, both during training of the linear classifier and at final evaluation, and it is denoted as (*Central*). An alternative evaluation is that an image embedding is generated by inputting a certain number of patches (same scale as training time and upsampled) into the neural network and aggregating the patch embeddings by averaging. The number of aggregated patches is marked in the Figures and Tables. In Figure 2, we show that linear probing and k-NN evaluations are consistent. In the following, we mainly present linear probing evaluation. The results on CIFAR-10 and CIFAR-100 are shown in Tables 1 and 2 respectively. The main observation is that pretraining on small patches and evaluating with the averaged embedding performs on par or better than the baseline methods, as highlighted. The central crop evaluation performs significantly worse for patch-based training is expected since the network has never seen a whole image during training.

Table 2: **Performance on CIFAR-100 with patch-based and standard multi-scale SSL pre-training.** We evaluate the performance of a linear classifier with BagSSL and baseline SSL methods. BagSSL uses *patch-based training*, where $14 \times 14$ image patches are sampled during pretraining. Baseline methods use *multi-scale training*, where the patch scale is uniformly sampled between scale 0.08 and 1.0, which means that multi-scale crops are sampled uniformly from $9 \times 9$ to $32 \times 32$ patches.

| Method | *Patch-based training evaluation* | | | | *Multi-scale training evaluation* | | | |
|---|---|---|---|---|---|---|---|---|
| | Central crop | 1 patch | 16 patches | 256 patches | Central crop | 1 crop | 16 crops | 256 crops |
| SimCLR | 34.7 | 59.4 | 67.2 | 67.4 | 66.8 | 60.5 | 68.2 | 68.3 |
| TCR | 34.6 | 59.2 | 67.1 | 67.3 | 66.8 | 60.5 | 68.1 | 68.3 |
| VICReg | 35.5 | 60.1 | 68.0 | 68.3 | 67.6 | 61.4 | 69.0 | 69.3 |
| BYOL | 37.4 | 60.9 | 68.9 | 69.2 | 68.8 | 62.3 | 69.7 | 69.9 |

Table 3: **Performance on ImageNet-100 with patch-based and standard multi-scale SSL pre-training.** We evaluate the performance of a linear classifier with BagSSL and baseline SSL methods. BagSSL uses *patch-based training*, where $100 \times 100$ patches are sampled during pretraining. Baseline methods use *multi-scale training*, where the patch scale is uniformly sampled between scale 0.08 and 1.0, which means that multi-scale crops are sampled uniformly from $64 \times 64$ to $224 \times 224$ patches.

| Method | *Patch-based training evaluation* | | | | *Multi-scale training evaluation* | | | |
|---|---|---|---|---|---|---|---|---|
| | Central crop | 1 patch | 16 patches | 48 patches | Central crop | 1 crop | 16 crops | 48 crops |
| SimCLR | 41.4 | 45.7 | 76.1 | 76.2 | 77.5 | 70.3 | 78.6 | 79.0 |
| TCR | 41.3 | 45.6 | 76.1 | 76.3 | 77.3 | 70.1 | 78.5 | 78.8 |
| VICReg | 42.1 | 46.1 | 76.8 | 76.9 | 77.8 | 70.7 | 79.1 | 79.4 |
| BYOL | 42.9 | 47.3 | 77.9 | 77.7 | 78.0 | 71.1 | 79.4 | 80.1 |

**ImageNet.** We also provide experimental results on ImageNet-100 and ImageNet datasets Deng et al. (2009) with ResNet-50 and linear probing evaluation protocol. ImageNet-100 and ImageNet results are shown in Table 3 and Figure 3(b) respectively. The behavior observed on CIFAR-10 generalizes to ImageNet-100. Averaging embeddings of small patches produced by the patch-based pretrained models perform comparably to standard "central" evaluation of the embedding produced by the baseline models, as highlighted in Table 3. In Figure 3(b), we show the results on ImageNet. The patch-based pretrained model achieves 67.9% top-1 accuracy with 16 patch embedding averaged, whereas the baseline method VICReg achieves 73.2% top-1 accuracy. This 5.3% performance gap might be due to sub-optimal hyperparameters as we did not optimize the hyperparameters for patch-based training pretraining. Interestingly, with $32 \times 32$ patch representation, BagSSL achieves 62% top-1 linear probing accuracy.

## 4.2 Patch Embedding Aggregation Approximately Converges to the Whole-Image Embedding

In this experiment, we show that for a multi-scale pretrained baseline SSL model, VICReg, linearly aggregating the patch embedding approximately converges to the whole-image embedding. This baseline VICReg network checkpoint has been trained on image crops ranging from $64 \times 64$ to $224 \times 224$. We randomly select 512 images from the ImageNet dataset. For each image, we first get the $224 \times 224$ center crop embedding. We also average embeddings of $N$ random $100 \times 100$ patches for each image. Then we calculate the cosine similarity between the patch-aggregated and center-crop embedding. Figure 3(a) shows that the aggregated embedding approximately converges to the whole-image embedding as $N$ increases from 1 to 16 to all the image patches[1].

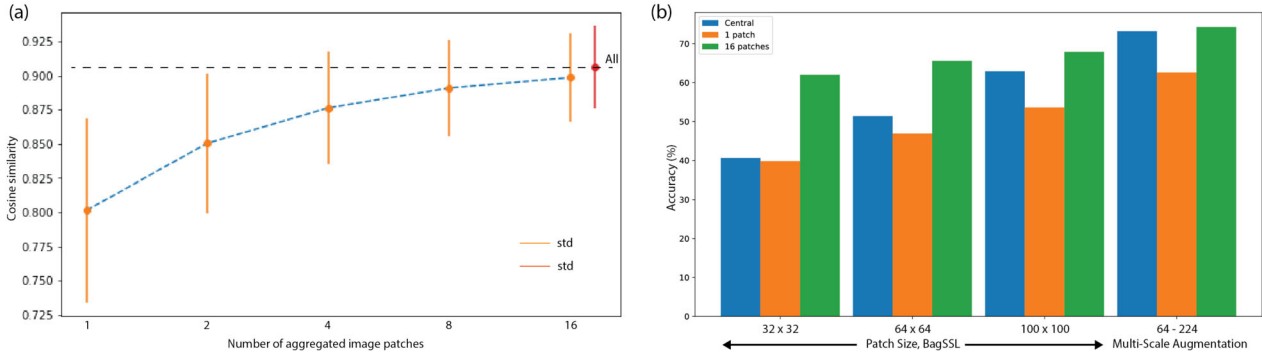

Figure 3: **(a) Patch embedding approximate convergence to the instance embedding.** For a baseline multi-scale pretrained VICReg model, we show that the patch embedding aggregation approximately converges to the whole-image embedding as the number of aggregated patches increases. We evaluate the cosine similarity between the aggregation of $N$ patch embeddings and the whole-image embedding. $N$ is selected from $1, 2, 4, 8, 16$ and all possible patches in the image. **(b) Linear evaluation on ImageNet for various `RandomResizedCrop` scales.** We show the performance of a linear classifier for various pretraining and evaluation settings. BagSSL uses fixed-scale patches, and the baseline method, VICReg, uses multi-scale augmentation `RandomResizedCrop(0.08, 1.0)`, which corresponds to randomly and uniformly select crop scale ranging from $64 \times 64$ to $224 \times 224$. For each pretraining setting, we provide three different evaluations: 1) "Central": the standard evaluation, which takes a $224 \times 224$ crop; 2) "1 patch": this evaluation takes 1 patch or crop following the corresponding pretraining setting; 3) "16 patches": this evaluation takes 16 randomly selected patches or crops following the corresponding pretraining setting.

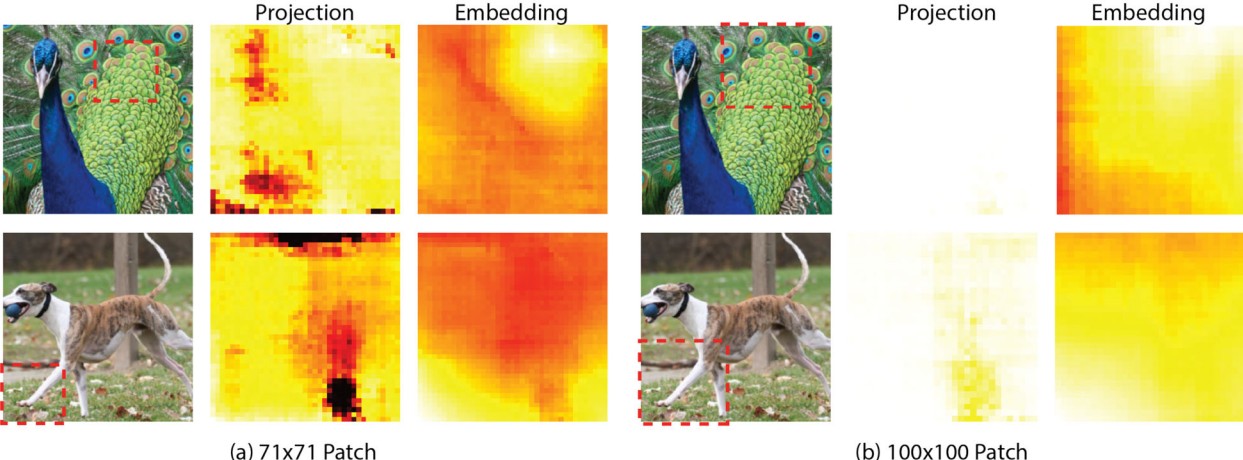

Figure 4: **Visualization of cosine similarity in the projection space and the embedding space.** Query patch is indicated by red dash. Projection and Embedding cosine-similarity heatmaps use the same color scaling. The projection vectors are significantly more invariant compared to the embedding ones, and the embedding space contains localized information that is shared among similar patches, when the size of the patches is small enough. We can see that the embedding space tends to preserve more locality compared to the projection space.

## 4.3 Patch Representation Preserves Locality

Joint-embedding SSL methods are primarily motivated from an invariance perspective. While this perspective is relatively accurate globally, we show that locality is preserved when we zoom into local patch-level representation. In this section, we provide nearest neighbor visualization on CIFAR-10 and cosine-similarity

---

[1]"All": extracting overlapped patches with stride 4 and totally aggregate about 1000 patches' embeddings per image.

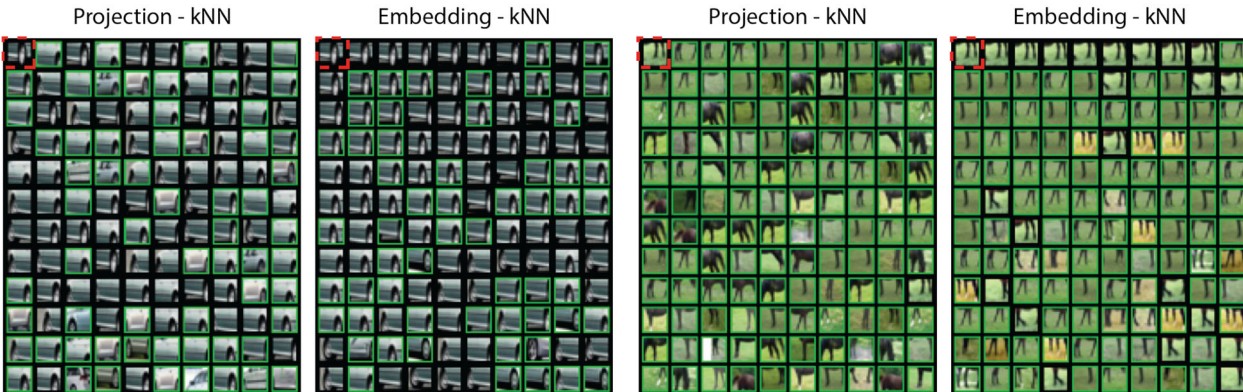

Figure 5: **Visualization of kNN in the projection space and the embedding space for CIFAR10.** Distance is calculated by cosine similarity. The query patch is in the top left corner encircled by red-dash box, the green box indicates patches from other images of the same class. Patches without surrounding box is from the same image as the query. While the nearest neighbors are both from same-category instances, we can see that the embedding space tends to preserve the local part information, whereas the projection space may collapse different parts of the same category.

heatmap visualization on ImageNet to show this point and provide a deeper understanding of the learned representation.

First, we take a baseline VICReg model pretrained on ImageNet, for a given image patch (e.g., circled by red dash boxes in Figure 4), we visualize the cosine-similarity between embedding from this patch and that from the other same-size patches from the same image. The heatmap visualization is normalized to the same scale. While the cosine-similarity heatmap for large ($100 \times 100$) patches is more invariant, as shown in Figure 4(b), the heatmap for smaller ($71 \times 71$) patches preserves the locality relatively well as demonstrated in Figure 4(a).

Next, we pre-train a model with only $14 \times 14$ image patches on CIFAR-10 and calculate the projection and embedding vectors of all different image patches from the training set. Then for a given $14 \times 14$ image patch (e.g., the ones circled by red dash boxes Fig 5), we visualize its $k$ nearest neighbors in terms of cosine-similarity in both the projection and the embedding space. Figure 5 shows the results for two image patches. The patches circled by green boxes are image patches from another image of the same category, whereas the uncircled patches are from the same image.

Overall, we observe that the projection vectors are significantly more invariant than the embedding vectors. This is apparent from Figure 4 and Figure 5. For the CIFAR kNN patches, NNs in the embedding space are visually much more similar than NNs in the projection space. In the embedding space, the nearest NNs are mostly locally shifted patches of similar "part" information. For projection space, however, many NNs are patches of different "part" information from the same class. E.g., we can see in Figure 5 that an NN of a "wheel" in the projection space might be a "door" or a "window". However, the NNs in the embedding space all contain "wheel" information. In the second example, the NNs of a "horse legs" patch may have different "horse" body parts, whereas the NNs in the embedding space are all "horse leg".

The heatmap visualization on ImageNet also illustrates the same phenomenon. The projection vector from a patch is highly similar to that from the query patch whenever the patch has enough information to infer the class of the image. For embedding vectors, the similarity area is much more localized to the query patch or other patches with similar features (the other leg of the dog in Figure 4). This general observation is consistent with the results of the visualizations in Bordes et al. (2022). A more thorough visualization is provided in Appendix E.

Table 4: **Linear evaluation with aggregated embedding on ImageNet with models trained with state-of-the-art SSL methods.** Using aggregated embedding outperforms embedding from the center crop. Central: Embedding from the center cropped image is used in training and testing using the standard linear evaluation protocol. 1, 16, and 48 crops: The linear classifier is trained and evaluated on the aggregated embedding of 1, 16, and 48 crops, respectively, sampled with the same scale factor range as during pretraining (0.08, 1.0).

| Method | Central crop | 1 crop | 16 crops | 48 crops |
|--------|--------------|--------|----------|----------|
| SimCLR | 69.3 | 54.3 | 71.0 | 71.3 |
| TCR | 69.0 | 54.1 | 70.7 | 71.1 |
| VICReg | 73.2 | 57.6 | 74.2 | 74.4 |
| BYOL | 74.3 | 59.3 | 75.4 | 75.6 |

Table 5: **Evaluation of SOTA SSL models and these models with linearly-aggregated patches embedding enhancement.** All the baseline SSL model uses ResNet-18 as the backbone. We apply spatial average pooling on the last layer output of ResNet-18 and treat it as feature. We evaluate the performance of these checkpoints with both a linear classifier and a K-nearest-neighbor (KNN) classifier. For the "Enhancement" evaluation, the KNN classifier is evaluated on the linearly-aggregated embedding of 25 patches with size $16 \times 16$. These patches are sampled using a sliding window with stride 4.

| Method | *Baseline (KNN)* | *Baseline (Linear)* | *Local Aggregation (KNN)* | *Local Aggregation (Linear)* |
|--------|------------------|---------------------|---------------------------|------------------------------|
| SimCLR | 90.2 | 90.7 | 93.1 | 92.7 |
| VICReg | 90.8 | 91.2 | 93.1 | 92.2 |
| BYOL | 91.5 | 92.6 | 93.5 | 94.0 |

## 4.4 Patch-Based Evaluation Enhances the SSL Baselines

**Global Aggregation.** The results in Section 4.1 show that the best performance is obtained when the pretraining step uses multi-scale crops and the evaluation step uses the aggregated patch embeddings. Here, we evaluate the patch-aggregated representation with four baseline methods pretrained on ImageNet to further confirm this performance boost. All models are downloaded from their original repository. Table 4 shows the linear evaluation performance on the validation set of ImageNet using the whole-image embedding and patch-aggregated embedding. For all these models, aggregated embedding outperforms whole-image evaluation, often by more than 1%. Also, increasing the number of patches averaged in the aggregation process improves the performance. We do not go beyond 48 patches due to engineering limits on memory and runtime. Still, we hypothesize that a further increase in the number of patches will improve the performance, as demonstrated on CIFAR-10, where 256 patches significantly outperform 16 patches.

**Local Aggregation.** In this work, the embedding aggregation is global or uses bag-of-local features. The drawback of doing so is that spatial information can be lost. An alternative way to aggregate patch embeddings is local averaging and concatenating locally averaged embeddings into a single long vector. Empirically, such a method tends to outperform the global average significantly. To show the results, we use the checkpoints of the SOTA SSL model pretrained on the CIFAR10 dataset from SoloLearn (da Costa et al., 2022) and tested linear and kNN accuracy with local concatenation aggregation. As shown in Table 5, concatenation aggregation further improve the performance of these SOTA SSL model. Even with only 25 patches, the k-NN accuracy of the aggregated embedding outperforms the baseline linear evaluation accuracy by a larger margin compared to global aggregation results shown in Table 1.

## 5   Discussion

In this paper, we seek to understand the success of joint-embedding SSL methods. We establish a formal connection between joint-embedding SSL and the co-occurrence of image patches. We demonstrate learning an embedding for fixed-size image patches and linear aggregating the local patch embeddings can achieve similar or even better performance than the baseline models pretrained with multi-scale crops. On the other hand, with a multi-scale pretrained model, we show that the whole image embedding is approximately the average of local patch embeddings. Given this, invariance is expected at the global scale. Through visualization, we show that the locality is preserved when we zoom into local patch-level representation. These findings supplement the prevailing invariance perspective and show that there is a distributed representation of local image patches behind the success of joint-embedding SSL. The insights from this angle also help us enhance the representation quality from the baseline methods.

During training, the computational cost for BagSSL and the corresponding baseline is almost exactly the same. During inference, the computational cost for BagSSL is $N$ time larger than the baseline (central crop eval), where $N$ is the number of aggregated patches. Although the eval computation is much higher, this disadvantage is somewhat tangential to the goal of this work, which is to provide an understanding of self-supervised learning. One might question whether BagSSL increases the representation capacity by having much computation during evaluation. Though BagSSL has much more computation during evaluation, it does not necessarily have much more capacity since we linearly aggregate the patch representation, which is essentially a bag-of-local-feature approach. For example, Table 1 shows that for multi-scale training evaluation, linear aggregating of the crop representations outperforms the default evaluation for the baseline methods. However, the capacity improvement is limited.

There are a few limitations to this work. First, we still use the commonly used projector head, whose role is yet to be fully understood. Second, we adopted the convention of two-crop training for BagSSL. If joint-embedding SSL aims to model the co-occurrence of patches, two local patches would be relatively inefficient and create a disadvantage during training. Third, we haven't optimized the hyperparameters thoroughly for BagSSL on ImageNet-1K due to engineering resource limitations. Fourth, the results from this work only provides a potential explanation to the SSL baselines, and it does not decisively show that the SSL methods is only learning a patch-based representation. Given the second and third limitations, BagSSL results may be further improved, given better engineering. We leave these to our future work.

### Acknowledgments

We thank Zeyu Yun for helping us with the local aggregation evaluation and providing feedback during the preparation of this paper. We thank many colleagues from CCN/CCM@Flatiron Institute for the valuable comments.

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
