# OpenReview forum: "Bag of Image Patch Embedding Behind the Success of Self-Supervised Learning"
_TMLR — Accepted by TMLR_

### Review · Reviewer_QgnZ · 2023-07-20

**Summary Of Contributions:**

This paper challenges the traditional understanding of Self-Supervised Learning (SSL) for image representation, proposing that joint-embedding SSL primarily learns through the co-occurrence of image patches. Introducing the concept of Bag of Self-Supervised Learning (BagSSL), the authors demonstrate improved or comparable results to baseline methods by aggregating local patch representations. They further show that, in a multi-scale pre-trained model, the whole image embedding can be approximated as the average of these local patch embeddings. Evidence is provided that this approach enhances various state-of-the-art baseline methods. The paper argues that patch representation simplifies the understanding of SSL and helps demystify self-supervised representation learning.

**Audience:**

Yes

**Broader Impact Concerns:**

No concerns.

**Claims And Evidence:**

Yes

**Requested Changes:**

**Requests**

1. Discussion about computational cost
2. Comparison with crop aggregation in Tables 4 and 5

**Strengths And Weaknesses:**

**Strengths.**

- **Intuitive approach.** The introduced approach in defining SSL through the lens of co-occurrence of image patches provides a fresh and insightful perspective, potentially reshaping the understanding and application of SSL in image representation.

**Weaknesses.**

- **Computational cost.** While the authors have focused on the performance improvement from patch aggregation, they do not report the increase of computational cost due to the multiple inferences of each patch. To fairly argue the benefit of patch aggregation compared to Central crop (Tables 1 to 3), it should be discussed.
- **Lack of important baselines.** In Section 4.4, the authors further demonstrate the usefulness of patch embeddings by showing that their aggregation can boost the performance of baseline SSL. However, the similar performance boosting with aggregation can be considered without patch. As the authors already have shown in Tables 1 to 3, crop-based aggregation might be able to outperform patch-based aggregation. But, such results are not provided in Tables 4 and 5.
- **Different messages between Section 4.1 and 4.2.** The authors have denoted that “averaged embedding performs on par or better than baselines (Central)” in Section 4.1. Also, the improvement is enlarged as the number of patches is increased. At the same time, in Section 4.2, they have also denoted that aggregated embedding converges to the whole-image embedding, which is obtained with center crop. Then, does it mean that the performance of patch aggregation is increased until the specific number of patches, then it start to decreased and converge to performance of central crop?

---

> ### Author Response · Authors · 2023-08-25
> **Reply to Reviewer QgnZ:**
>
> “Computational cost.”
>
> This is a very useful suggestion! As mentioned in the general response, a dedicated discussion about computational cost is added to the revision. During training, the computational cost for BagSSL and the corresponding baseline is almost exactly the same. During inference, the computational cost for BagSSL is $N$ time larger than the baseline (central crop eval), where $N$ is the number of aggregated patches. Although the eval computation is much higher, this disadvantage is somewhat tangential to the goal of this work, which is to provide an understanding of self-supervised learning. However, we agree that adding such a discussion can potentially clarify some related confusion.
>
> “As the authors already have shown in Tables 1 to 3, crop-based aggregation might be able to outperform patch-based aggregation. But, such results are not provided in Tables 4 and 5.”
>
> Good catch! Table 4 and Table 5 are exactly dedicated to showing this point: training with multi-scale crops and evaluation with different aggregation settings of multi-scale crops. However, we have slightly abused the term and used “patches”, which was dedicated to describing fixed-scale image patches. If the review read the caption, it would clarify this confusion. We have revised the words in Table 4 accordingly. Please kindly refer to the corresponding section in the revision. In this work, the embedding aggregation is global or uses bag-of-local features. The drawback of doing so is that spatial information can be lost. Table 5 shows an alternative way to aggregate the embeddings beyond bag-of-local features, which provides better performance than global linear aggregation.
>
> “Different messages between Section 4.1 and 4.2.”
>
> Good point! We have added “approximately” to every convergence statement in the paper to make it more precise and consistent.

---

### Review · Reviewer_5Jcd · 2023-07-21

**Summary Of Contributions:**

This paper proposes a simple technique that learns image patch embeddings and aggregates local patch embeddings as image-level representations. The authors named the proposed method as BadSSL. Empirically the paper shows that the proposed method improves baseline self-supervised learning methods. Also, the authors prove that the spectral contrastive loss function is equivalent to co-occurence statistics modeling loss function.

**Audience:**

Yes

**Broader Impact Concerns:**

This work has no broader impact concern.

**Claims And Evidence:**

Yes

**Requested Changes:**

* Table 5. Does Local Aggregation improve linear as well? The performance of baselines is reported using both KNN and linear. But the performance of the proposed method/Local aggregation with linear was not reported. If the results are included, it will be more comprehensive and fairer.
* Computational costs. Since the proposed method uses many patches, it might require much larger computing resources. But. the computational costs and efficiency are not explicitly compared/discussed.
* Does the proposed method use the same data augmentation? It can be viewed as a type of data augmentation.

**Strengths And Weaknesses:**

Strengths
----

1. The proposed method is simple and effective. It improves the performance of various self-supervised learning baseline algorithms.
2. The paper discusses the relationship between spectral contrastive loss and co-occurrence statistics loss.
3. The analysis of aggregated patch embedding shows that it converges to the image-level representation of baseline algorithms.
4. The effectiveness of the proposed method was demonstrated in both single-scale and multi-scale augmentation settings.

Weaknesses
----

1. The presentation of this paper, especially the writing, needs to be improved. It does not read well.
2. Table 1,2,3.  do not directly compare with baseline algorithms. For instance, Table 1 shows CIFAR-100 performance. However, the performance is significantly lower than the baselines. SimCLR achieved about 80.2 in the original paper. But Table 2 the proposed method with SimCLR achieved 67.4 in Patch-based training evaluation and 68.3 in multi-scale training evaluation. The authors need to provide experimental results that are compatible with the original self-supervised papers.
3. Considering the simplicity of the proposed method, the method should provide strong empirical/practical gain. But as discussed above, the proposed method shows poor performance.
4. Compared to Multi-scale training, the gain from patch-based training is relatively small.

---

> ### Author Response · Authors · 2023-08-25
> **Reply to Reviewer 5Jcd**
>
> “Table 5. Table 5. Does Local Aggregation improve linear as well? .. If the results are included, it will be more comprehensive and fairer.”
>
> Good suggestion! The following results are the updated Table 5 included in the revision. And it shows that local aggregation improves linear probing as well.
>
> | Method | Baseline (KNN) | Baseline (Linear) | Local Aggregation (KNN) | Local Aggregation (Linear) |
> | :-----------:|  :-----------: | :-----------: | :-----------: | :-----------: |
> | SimCLR |    90.2 % | 90.7 % | 93.1 % | 92.7 % |
> | VICReg | 90.8 % | 91.2 % | 93.1 % | 92.2 % |
> | BYOL | 91.5 % | 92.6 % | 93.5 % | 94.0 % |
>
> “Computational costs. Since the proposed method uses many patches, it might require much larger computing resources. But. the computational costs and efficiency are not explicitly compared/discussed.”
>
> This is also a very useful suggestion! As mentioned in the general response, a dedicated discussion about computational cost is added to the revision. During training, the computational cost for BagSSL and the corresponding baseline is almost exactly the same. During inference, the computational cost for BagSSL is $N$ time larger than the baseline (central crop eval), where $N$ is the number of aggregated patches. Although the eval computation is much higher, this disadvantage is somewhat tangential to the goal of this work, which is to provide an understanding of self-supervised learning. However, we agree that adding such a discussion can potentially clarify some related confusion.
>
> “Does the proposed method use the same data augmentation? It can be viewed as a type of data augmentation.”
>
> If we understand this question correctly, yes. During training, the only difference between BagSSL and the baseline is that BagSSL uses fixed-scale image patches as crops, and the baseline uses multi-scale crops. And yes, extracting fixed-scale image patches from each sample can be considered a type of augmentation.
>
> “Table 1,2,3. do not directly compare with baseline algorithms. For instance, Table 1 shows CIFAR-100 performance. However, the performance is significantly lower than the baselines. SimCLR achieved about $80.2$ in the original paper. But Table 2 the proposed method with SimCLR achieved $67.4$ in Patch-based training evaluation and $68.3$ in multi-scale training evaluation. The authors need to provide experimental results that are compatible with the original self-supervised papers.”
>
> The baseline results provided in this work are relatively competitive and reliable. If we understand the reviewer correctly, the $ 80.2 $% result comes from Table 8 in SimCLR paper. The experiment setting is very different. If the reviewer reads the caption in SimCLR Table 8 carefully, it is a transfer learning experiment, where a ResNet 50-4X is pretrained on ImageNet and transferred to CIFAR100 with linear probing. In our CIFAR100 baseline experiment (Table 2), we pre-train a ResNet-34 only on CIFAR100 with SoloLearn (https://github.com/vturrisi/solo-learn), which is reliable and widely adopted by many recently published works.
>
> “The presentation of this paper, especially the writing, needs to be improved. It does not read well.”
>
> We have fixed many typos and grammar issues, revised some precise statements, and added new requested results in the revision. Most of these changes are marked red, and the removed parts are marked with strike-out. If the reviewer has more specific suggestions, we will happily address the issues in the next revision.
>
>
> “Compared to Multi-scale training, the gain from patch-based training is relatively small.”
>
> There shall not be any gain from patch-based training. The main point of this work is to provide understanding and show that a patch-based representation can explain the performance of self-supervised learning and provide further understanding. The linear aggregation and local aggregation evaluations improve the performance, which is a relatively minor point of this work.
>
> “.. the authors named the proposed method as BadSSL”
>
> This must be a typo .. it is named BagSSL.

---

### Review · Reviewer_tLRJ · 2023-08-02

**Summary Of Contributions:**

The paper presents a patch-based SSL method named BagSSL. The idea is to learn embeddings of image patches with SSL where patches from the same image are considered positive. They also provide a patch aggregation method (mean of patch embeddings) to obtain image embedding. Results are provided on cifar-10/100, and imagenet-100/1k with SimCLR, TCR, VICReg and BYOL methods.

**Audience:**

Yes

**Claims And Evidence:**

No

**Requested Changes:**

1. Clear justification for the main claims would strengthen the paper and position it correctly in the literature.

**Strengths And Weaknesses:**

## Strengths

1. The patch-based SSL idea is interesting and it is competitive compared to the standard SSL approach which uses multi-scale crop augmentation.
2. The patch aggregation method improves even the baseline SSL methods which were trained with multi-scale augmentations.
3. Overall the writing is clear.

## Weaknesses

1. There are unsubstantiated claims.
a. SSL learns a representation of patches (abstract and introduction) is unsubstantiated as the paper doesn't show it without a doubt in my opinion. The BagSSL method performing similarly or patch aggregation improving SSL methods does not justify this claim.
b. Proposition 3.1 only shows that patch-based SSL loss is similar to co-occurrence loss and it doesn't show this for a general SSL which uses full image embeddings.
c. Sec 4.1 claims the performance difference between BagSSL and VICReg is due to engineering issues or suboptimal hyperparameters which is unsubstantiated.

2. BagSSL learns embeddings of patches and therefore, an image embedding is obtained by patch aggregation (ie, an image corresponds to a set of patch embeddings). One could think about this as BagSSL having more capacity to embed an image. Nevertheless, BagSSL is inferior to multi-scale crop augmentation with baseline SSL. Can you comment on this?

---

> ### Author Response · Authors · 2023-08-25
> **Reply to Reviewer tLRJ, Part I**
>
> “a. SSL learns a representation of patches is unsubstantiated .. BagSSL method performing similarly .. does not justify this claim.”
>
> A representation is a transform that turns a raw signal into a new space such that the structures are explicit, which is measured by benchmark probing accuracy. Whether the baseline representation leverages more than patch-level structures to form the representation depends on whether the baseline representation makes the structures more explicit than purely using patch-level representation. We show that linear aggregating patch-level representation leads to similar performance. The baseline methods are rather an efficient aggregation of patch-level representation.
>
> “b. Proposition 3.1 only shows that patch-based SSL loss is similar to co-occurrence loss and it doesn't show this for a general SSL which uses full image embeddings.”
>
> Proposition 3.1 also shows that a general SSL with multi-scale augmentation is equivalent to the co-occurrence modeling of multi-scale crops. Essentially, one can treat p(x_1,x_2) as the joint distribution of two co-occurring multi-scale crops. The argument in Proposition 3.1 does not depend on a fixed-scale patch assumption. Together with Garrido et al. 2023, Proposition shows that contrastive and non-contrastive SSL methods are modeling the co-occurrence of multi-scale image crops. Since modeling fixed-scale image patches gives a similar representation performance, we can conclude that joint-embedding SSL methods are modeling the co-occurrence of fixed-scale image patches.
>
> “c. Sec 4.1 claims the performance difference between BagSSL and VICReg is due to engineering issues or suboptimal hyperparameters which is unsubstantiated.”
>
> In the statement, we stated the following: “.. gap might be due to sub-optimal hyperparameters as we did not optimize the hyperparameters for patch-based training pretraining.” This is our educated guess, given our extensive experience working on this benchmark.

---

> > ### Comment · Reviewer_tLRJ · 2023-10-20
> > **Response**
> >
> > Thank you for the detailed experiments and response.
> >
> > Regarding (a) there could be many reasons that the two approaches give similar results, in that case one cannot claim that both approaches are the essentially doing the same thing under the hood. Deeper analysis is needed to verify this without a doubt. Having said that, it is a high-level comment and it is not always possible to do this especially with deep learning. Therefore, I would suggest to reword the claims to make sure they match the empirical observations.

---

> > > ### Author Response · Authors · 2023-11-12
> > > **Revision**
> > >
> > > "Regarding (a) .. "
> > > We agree that none of the results in the paper decisively show that the baseline methods are only learning a patch-based representation. The evidence from different angle only provide a potential explanation to the underlying representation. We have gone through the paper and adjusted every claim that we feel may be too strong. And the submitted revision shall be reflecting the evidence more properly.
> > >
> > > Thanks for the additional suggestion!

---

> ### Author Response · Authors · 2023-08-25
> **Reply to Reviewer tLRJ, Part II**
>
> “.. One could think about this as BagSSL having more capacity to embed an image. Nevertheless, BagSSL is inferior to multi-scale crop augmentation with baseline SSL. Can you comment on this?”
>
> This is a great point!
> First, BagSSL has much more computation during evaluation, as discussed in the general reply. But BagSSL does not necessarily have much more capacity since we linearly aggregate the patch representation, which is essentially a bag-of-feature approach. For example, in Table 1 for multi-scale training evaluation:
>
> | Method | Central crop | 256 crops |
> | :-----------:| :-----------: | :-----------: |
> | SimCLR|    90.2 %  |    91.8 %  |
> | TCR | 90.1 % |  91.8 % |
> | VICReg | 90.7 %| 92.0% |
> | BYOL | 90.9 %| 92.4 % |
>
> As one can see, linear aggregating of the crop representations does outperform the default evaluation for the baseline methods. However, the capacity improvement is limited.
>
> Second, BagSSL outperforms the baseline method (w/ default evaluation) on the CIFAR10 and CIFAR100, as shown in Table 1 and Table 2. BagSSL slightly underperforms the baseline method (w/ default evaluation) on ImageNet-100 and ImageNet-1K, as shown in Table 3 and Figure 3. So, BagSSL is not always inferior to the multi-scale crop augmentation-trained baseline SSL methods. If we insist on an absolute apple-to-apple comparison, the following tables might be useful:
>
> CIFAR10
> | Method | BagSSL 256-patches | Multi-scale 256-crops | Multi-scale Central-crop |
> | :-----------: | :-----------: | :-----------: | :-----------: |
> | SimCLR |    90.8 %  |    91.8 %  | 90.2 % |
> | TCR | 90.8 % |  91.8 % | 90.1 % |
> | VICReg | 91.2 %| 92.0% | 90.7 % |
> | BYOL | 91.5 %| 92.4 % | 90.9% |
>
>
> CIFAR100
> | Method | BagSSL 256-patches| Multi-scale 256-crops| Multi-scale Central-crop |
> | :-----------:|  :-----------: | :-----------: | :-----------: |
> | SimCLR |    67.4% | 68.3 % | 66.8 % |
> | TCR | 67.3 % | 68.3 % | 66.8 % |
> | VICReg | 68.3 % | 69.3 % | 67.6 % |
> | BYOL | 69.2 % | 69.9 % | 68.8 % |
>
>
> ImageNet-100
>
> | Method | BagSSL 48-patches| Multi-scale 48-crops| Multi-scale Central-crop |
> | :-----------:|  :-----------: | :-----------: | :-----------: |
> | SimCLR |    76.2 % | 79.0 % | 77.5 % |
> | TCR | 76.3 % | 78.8 % | 77.3 % |
> | VICReg | 76.9 % | 79.4 % | 77.8 % |
> | BYOL | 77.7 % | 80.1 % | 78.0 % |
>
> As we can see, fixed-scale patch training does underperform the multi-scale crop training slightly in all cases when we use linear aggregation evaluation.
>
> Third, BagSSL has a considerable sampling disadvantage. In both BagSSL and the baseline method, for each sample image, we extract two patches or crops. BagSSL uses only fixed-sized patches, while baseline methods can sample much larger crops. As we have mentioned, larger crop representation is approximately a linear aggregation of the smaller ones’ representation. When the baseline method uses a large crop, it can be considered the average of many smaller ones. This more or less works as a memory bank. So, baseline methods tend to have better sampling efficiency given the same amount of patch/crop augmentation per sample. However, the main purpose of this paper is to show that one can use only patch-level representation to achieve a similar performance.

---

### Author Response · Authors · 2023-08-25
**General Reply**

We thank our reviewers for their encouraging comments, helpful suggestions, and insightful questions. Reviewer1 tLRJ says, “.. idea is interesting and it is competitive ..”; Reviewer2 5Jcd says, “.. proposed method is simple and effective ..” Reviewer3 QgnZ says, “.. provides a fresh and insightful perspective, potentially reshaping the understanding and application of SSL ..” The questions and suggestions help us revise the paper to deliver key messages better. Both reviewer 5Jcd and reviewer QgnZ raised the question of computational cost. Reviewer tLRJ also raised the question of representation capacity. We address these questions first in this general reply. Then, we also provide an overview of the revision alongside a revised paper. The added parts are red in the revision, whereas strike lines mark the removed parts.

During training, the computational cost for BagSSL and the corresponding baseline is almost precisely the same. During inference, the computational cost for BagSSL is $N$ time larger than the baseline (central crop eval), where $N$ is the number of aggregated patches. Although the eval computation is much higher, this disadvantage is somewhat tangential to the goal of this work, which is to provide an understanding of self-supervised learning. However, we agree that adding such a discussion can potentially clarify some related confusion.

A representation transforms a raw signal into a new space such that the structures are explicit, which is measured by benchmark probing accuracy. Whether the baseline representation leverages more than patch-level structures to form the representation depends on whether the baseline representation makes the structures more explicit than purely using patch-level representation. We show that linear aggregating patch-level representation leads to similar performance. The baseline methods are rather an efficient aggregation of patch-level representation.

BagSSL has much more computation during evaluation. But BagSSL does not necessarily have much more capacity since we linearly aggregate the patch representation, which is essentially a bag-of-local-feature approach. For example, in Table 1 for multi-scale training evaluation:
| Method | Central crop | 256 crops |
| :-----------:| :-----------: | :-----------: |
| SimCLR|    90.2 %  |    91.8 %  |
| TCR | 90.1 % |  91.8 % |
| VICReg | 90.7 %| 92.0% |
| BYOL | 90.9 %| 92.4 % |

As one can see, linear aggregating of the crop representations does outperform the default evaluation for the baseline methods. However, the capacity improvement is limited.

Here, we provide an overview of the changes made in the revision:

1. Linear probing results are added to Table 5;
2. Fixed typos and grammar issues;
3. Fixed the wording confusion in Table 4 and made the convergence statements more precise in the text;
4. A discussion on computation and capacity is added to the end of the paper.

Next, we reply to each reviewer to address the specific questions.

---

### Decision · Action_Editor_h4Li · 2023-11-06

**Recommendation:** Accept with minor revision

**Comment:**

This paper shows that self-supervised learning (SSL) approaches based on Siamese network architecture could be interpreted as learning a representation of image patches, which reflects their co-occurrence. The reviewers originally had a number of concerns due to unsubstantiated claims and insufficient experiments. Through the discussion phasse, these concerns were mostly resolved; consequently, two Reviewers are leaning to accept while the one Reviewer (5Jcd) keeps the original score leaning to the reject. The idea of patch-level contrastive learning is not a new idea as pointed by Reviewer 5Jcd. However, adding a new perspective to analyze SSL methods based on patch-level co-occurrence is interesting. Overall, this is an interesting and useful paper; therefore, I recommend its acceptance. As in the response to author’s reply for Reviewer tlRJ, deeper analysis or rewording the claims should be included in the final draft.

**Audience:**

Yes

**Claims And Evidence:**

Yes

---

> ### Author Response · Authors · 2023-11-12
> **Camera Ready Version and Revision**
>
> We have submitted the revised camera ready version according to the TMLR format.
>
> Further, we have gone through the paper and adjusted all of the claims to make sure they are properly reflecting the evidence provided. We thank our reviewers and action editor for providing the insightful suggestions!